# Plasma Concentrations of Extracellular Vesicles Are Decreased in Patients with Post-Infarct Cardiac Remodelling

**DOI:** 10.3390/biology10020097

**Published:** 2021-01-30

**Authors:** Aleksandra Gąsecka, Kinga Pluta, Katarzyna Solarska, Bartłomiej Rydz, Ceren Eyileten, Marek Postula, Edwin van der Pol, Rienk Nieuwland, Monika Budnik, Janusz Kochanowski, Miłosz J. Jaguszewski, Łukasz Szarpak, Tomasz Mazurek, Agnieszka Kapłon-Cieślicka, Grzegorz Opolski, Krzysztof J. Filipiak

**Affiliations:** 11st Chair and Department of Cardiology, Medical University of Warsaw, 02-097 Warsaw, Poland; aleksandra.gasecka@wum.edu.pl (A.G.); plutakinga.01@gmail.com (K.P.); kat.solarska@gmail.com (K.S.); barteksg9@gmail.com (B.R.); moni2911@interia.pl (M.B.); kochanowski.janusz@gmail.com (J.K.); tmazurek@kardia.edu.pl (T.M.); grzegorz.opolski@wum.edu.pl (G.O.); krzysztof.filipiak@wum.edu.pl (K.J.F.); 2Laboratory of Experimental Clinical Chemistry, Amsterdam University Medical Center, University of Amsterdam, 1105 AZ Amsterdam, The Netherlands; e.vanderpol@amsterdamumc.nl (E.v.d.P.); r.nieuwland@amsterdamumc.nl (R.N.); 3Department of Experimental and Clinical Pharmacology, Centre for Preclinical Research and Technology, Medical University of Warsaw, 02-091 Warsaw, Poland; cereneyileten@gmail.com (C.E.); mpostula@wum.edu.pl (M.P.); 4Biomedical Engineering and Physics, Amsterdam University Medical Center, University of Amsterdam, 1105 AZ Amsterdam, The Netherlands; 51st Department of Cardiology, Medical University of Gdańsk, 80-952 Gdansk, Poland; jamilosz@gmail.com; 6Maria Sklodowska-Curie Bialystok Oncology Center, 15-027 Bialystok, Poland; lukasz.szarpak@gmail.com; 7Maria Sklodowska-Curie Medical Academy in Warsaw, 03-411 Warsaw, Poland

**Keywords:** heart failure, left-ventricular remodelling, acute myocardial infarction, extracellular vesicles, flow cytometry

## Abstract

**Simple Summary:**

A heart attack may lead to the remodelling of the cardiac muscle, which negatively affects patient’s prognosis. At present, the mechanisms of cardiac remodelling remain unclear. In patients with heart attack, many body cells become activated and release small particles, called extracellular vesicles, which can either aggravate cardiac injury, or contribute to healing of heart muscle. In our study, we hypothesized that the concentrations of these small particles in plasma allow to determine which patients will experience remodelling of the cardiac muscle after the heart attach. We found that concentrations of extracellular vesicles from endothelial cells, erythrocytes and platelets, measured directly the heart attack, were lower in patients who developed cardiac remodelling 6 months later, compared to patients who had no remodelling. Vesicles from endothelial cells and erythrocytes allowed to determine remodelling independently of other clinical features. Hence, decreased concentrations of these vesicles may on one hand be a sign of inappropriate cardiac repair mechanisms, and on the other hand may allow to identify patients, who will develop cardiac remodelling after the heart attack.

**Abstract:**

Background, the mechanisms underlying left ventricular remodelling (LVR) after acute myocardial infarction (AMI) remain obscure. In the course of AMI, blood cells and endothelial cells release extracellular vesicles (EVs). We hypothesized that changes in EV concentrations after AMI may underlie LVR. Methods, plasma concentrations of EVs from endothelial cells (CD146+), erythrocytes (CD235a+), leukocytes (CD45+), platelets (CD61+), activated platelets (P-selectin+), and EVs exposing phosphatidylserine after AMI were determined by flow cytometry in 55 patients with the first AMI. LVR was defined as an increase in left ventricular end-diastolic volume by 20% at 6 months after AMI, compared to baseline. Results, baseline concentrations of EVs from endothelial cells, erythrocytes and platelets were lower in patients who developed LVR (*p* ≤ 0.02 for all). Concentrations of EVs from endothelial cells and erythrocytes were independent LVR predictors (OR 8.2, CI 1.3–54.2 and OR 17.8, CI 2.3–138.6, respectively) in multivariate analysis. Combining the three EV subtypes allowed to predict LVR with 83% sensitivity and 87% specificity. Conclusions, decreased plasma concentrations of EVs from endothelial cells, erythrocytes and platelets predict LVR after AMI. Since EV release EVs contributes to cellular homeostasis by waste removal, decreased concentrations of EVs may indicate dysfunctional cardiac homeostasis after AMI, thus promoting LVR.

## 1. Introduction

Coronary artery disease (CAD) is a major public health problem, affecting around 126 million individuals worldwide [1]. Improvements in the pharmacological and interventional treatment of CAD, including primary percutaneous interventions (PCI) for acute myocardial infarction (AMI), decreased the short-term mortality rate after AMI [2,3]. Consequently, long-term complications including heart failure (HF) have become a leading cause of death in CAD [4].

After AMI, myocardial necrosis leads to formation of scar tissue. The structural and functional geometrical changes of the cardiac muscle are termed post-infarct left ventricular remodelling (LVR) [5]. Echocardiographic evaluation of LVR includes measurements of left ventricular end-diastolic and end-systolic volumes (LVEDV, LVESV), left ventricle ejection fraction (EF), and 3D assessment of left ventricle sphericity index [6]. The most widely used definition of LVR, applied also in this study, is a >20% increase in LVEDV at six months after AMI [7]. About 30% of the patients after anterior AMI and 17% after non-anterior AMI develop LVR, which increases the risk of HF and the mortality rate [7]. The pathophysiological mechanisms underlying LVR after AMI remain obscure. To prevent development of LVR in patients after AMI, beta-blockers, angiotensin-converting enzyme inhibitors or angiotensin receptor blockers and aldosterone antagonists are used [8]. However, since there are no reliable parameters to identify patients who will develop post-infarct LVR, the therapy cannot be tailored to the individual patients’ need. Hence, new pathophysiological insights and biomarkers for LVR after AMI are urgently required. Extracellular vesicles (EVs) are released by blood and vascular endothelial cells into the blood [9]. EVs are thought to transport biomolecules, including cytokines, signalling proteins and nucleic acids, between cells, and therefore are believed to be involved in intercellular communication [10]. Depending on the molecular cargo, EVs can affect both physiological and pathological processes, such as immune responses, angiogenesis and wound healing [11]. Due to the multiple roles of EVs in health and disease, EVs may provide new and potentially non-invasive biomarkers [12]. Therefore, we hypothesized that (i) the concentrations of EVs evaluated after AMI differ between patients with and without LVR after 6 months, and, if so, (ii) the changes in EV concentrations may underlie LVR after AMI.

We compared the concentrations of EVs after AMI between patients who developed LVR at 6 months, and those who did not, and evaluated the predictive value of EVs for LVR.

## 2. Materials and Methods

### 2.1. Study Design

We prospectively evaluated the predictive value of EVs for LVR in all patients participating in the Antiplatelet Therapy Effect on Extracellular Vesicles (AFFECT EV) Echocardiography Substudy [13]. AFFECT EV was an investigator-initiated, prospective study conducted at the 1st Chair and Department of Cardiology, Medical University of Warsaw, Poland in collaboration with the Vesicle Observation Centre, Amsterdam University Medical Centers (UMC), The Netherlands [14]. The study protocol, designed in compliance with the Declaration of Helsinki, was approved by the Ethics Committee of Medical University of Warsaw, approval number KB/112/2016, registered in the Clinical Trials database as NCT02931045, and published previously [13]. All participants provided written informed consent.

### 2.2. Study Participants

Study inclusion and exclusion criteria are listed in Appendix A. Patients were eligible for enrolment if they were (a) were admitted to the hospital due to the first ST-segment elevation of acute myocardial infarction (STEMI) or non-STEMI (NSTEMI) with an onset of symptoms during the previous 24 h, and (b) underwent PCI with stent implantation. STEMI was defined as persistent ST-segment elevation of at least 0.1 mV in at least two contiguous electrocardiography leads, or a new left bundle-branch block [15]. NSTEMI was diagnosed in patients presenting with typical anginal chest pain along with an elevation of cardiac troponin concentration in the peripheral blood and ST-segment changes (ST depression, transient ST elevation, T-wave changes) on electrocardiogram [16].

### 2.3. Treatment

All patients received standard treatment after AMI according to the guidelines, including double antiplatelet therapy, β-blocker, angiotensin-converting enzyme inhibitor or angiotensin receptor blocker, aldosterone receptor antagonist and protein pump inhibitor [15,16].

### 2.4. Clinical Data Collection

Data collected at baseline included demographics (age, gender), body mass index, initial diagnosis and cardiovascular risk factors, including arterial hypertension, diabetes, hyperlipidaemia, and smoking. In addition, routine laboratory parameters were recorded. Each patient underwent transthoracic echocardiography within 24 h after AMI and at 6-month follow-up visit. LVR was defined as an increase in LVEDV by 20% at 6 months after AMI, compared to baseline echocardiography.

### 2.5. Blood Collection and Handling

Peripheral venous blood samples were collected from fasting patients at a single time-point (within 24 h after AMI). With fasting, we mean ≥8 h after last consumption. Blood was collected and processed according to the recent guidelines to study EVs [17]. Briefly, blood was collected in 10 mL 0.109 mol/L citrated plastic tubes (S-Monovette, Sarstedt) via antecubital vein puncture using a 19-gauge needle, without tourniquet. The first 2 mL were discarded to avoid pre-activation of platelets. Within maximum 15 min from blood collection, platelet-depleted plasma was prepared by double centrifugation using a Rotina 380 R equipped with a swing-out rotor and a radius of 155 mm (Hettich Zentrifugen, Tuttlingen, Germany). The centrifugation parameters were: 2500 g, 15 min, 20 °C, acceleration speed 1, no brake. The first centrifugation step was done with 10 mL whole blood collection tubes. Supernatant was collected 10 mm above the buffy coat. The second centrifugation step was done with 3.5 mL plasma in 15 mL polypropylene centrifuge tubes (Greiner Bio-One B.V). Supernatant (platelet-depleted plasma) was collected 5 mm above the buffy coat, transferred into 5 mL polypropylene centrifuge tubes (Greiner Bio-One B.V., Vilvoorde, Belgium), mixed by pipetting, transferred to 1.5 mL low-protein binding Eppendorfs (Thermo Fisher Scientific, MA, USA), and stored in −80 °C until analyzed. Prior to analysis, samples were thawed for 1 min in a water bath at 37 °C.

### 2.6. Flow Cytometry

Concentration of EVs were measured by flow cytometry (A60-Micro, Apogee Flow Systems, Hertfordshire, UK). We diluted samples 2-fold to 1500-fold in in citrated (0.32%) phosphate-buffered saline (PBS) to achieve a count rate of less than 3000 events/s to prevent swarm detection [18]. Diluted samples were measured during 120 s at a flow rate of 3.01 μL per min. The trigger threshold was set at 14 arbitrary units of the side scatter detector, which corresponds to a side scattering cross section of 10 nm^2^. The reported concentrations describe the number of particles (a) that exceed the side scatter threshold, (b) have a diameter >200 nm as determined by Flow-SR [19], (c) have a refractive index <1.42 to exclude positively labelled chylomicrons [20], and (d) that are positive at the fluorescence detector(s) corresponding to the used label(s), per mL of platelet-depleted plasma. We measured concentrations of EVs from endothelial cells (CD146^+^), erythrocytes (CD235a^+^), leukocytes (CD45^+^), platelets (CD61^+^), activated platelets (P-selectin^+^), and EVs exposing phosphatidylserine (PS). Although a generic EV marker is lacking, proteins binding PS are commonly used to stain ~50% of all plasma EVs [21]. Hence, we used lactadherin to stain all (PS-exposing) plasma EVs. To improve the reproducibility of our EV flow cytometry experiments, we (a) applied the new reporting framework for the standardized reporting of EV flow cytometry experiments (MIFlowCyt-EV) [22], (b) calibrated all detectors, (c) determined the EV diameter and refractive index by the flow cytometry scatter ratio (Flow-SR) [19], and (d) applied custom-built software to fully automate data calibration and processing.

### 2.7. End-Points

The primary end-point was the difference between the concentrations of EVs after AMI in patients with and without LVR at 6 months. The secondary end-point was the predictive value of EVs for LVR at 6 months.

### 2.8. Statistical Analysis

The power calculation was based on the systematic review with meta-analysis of seven clinical studies, which demonstrated that plasma concentrations of EVs are two-fold higher in patients with acute coronary syndrome, compared to healthy controls [23]. About 30% of patients after AMI develops LVR. Sample size was calculated based on the following assumptions: (i) significance level for two-sided testing 0.05, (ii) test power 0.9, (iii) standard deviation (SD) +/− 1.5, (iv) estimated difference in mean EV concentrations between the group with and without AMI 2. Based on the above assumptions, the study should include at least 13 patients who will develop LVR (25% of the study group). Based on this sample size estimation, a total of 52 patients should be enrolled in the study. Assuming 10% of patients lost to follow-up, 60 patients were eventually included in the study.

Statistical analysis was conducted using IBM SPSS Statistics, version 24.0 (IBM). Categorical variables were presented as number and percentage and compared using Fischer’s exact test. A Shapiro–Wilk test was used to assess normal distribution of continuous variables. Continuous variables were presented as mean and SD or median with interquartile range and compared using an unpaired t-test or Mann-Whitney U test. The diagnostic ability of EVs to discriminate between patients with and without LVR and the cut-offs were calculated using a receiver operating characteristic (ROC) curve. Logistic regression model incorporating the subtypes of EVs with significant sensitivity and specificity (area under the ROC curve, AUC) and clinical characteristics were used to determine the best model for LVR prediction. Mortality and other adverse events were reported descriptively. A p-value below 0.05 was considered significant.

## 3. Results

The study flow and exclusion and inclusion overview are shown in Figure 1. Between January 2017 and June 2018, 60 patients were enrolled, and 55 patients were included in the final analysis (5 patients withdrew consent and did not attend the follow-up visit at 6 months). Patient characteristics are shown in Table 1. Twelve patients developed LVR at 6 months (22%). Cardiovascular risk factors and laboratory characteristics including haemoglobin, platelet count, troponin-I and C-reactive protein concentration at baseline were comparable between the groups. STEMI occurred more frequently in patients who developed LVR compared to patients without LVR, but this difference did not reach statistical significance (92% vs. 72%, *p* = 0.26). Echocardiography parameters and pharmacotherapy were comparable between the groups. All patients received dual antiplatelet therapy, all patients except for one received atorvastatin, and more than 80% of patients received a β-blocker, an angiotensin-converting enzyme inhibitor, and a proton pump inhibitor. At 6 months, LVEDV and LVESV were larger with patients with LVR, compared to those without LVR (*p* = 0.03, *p* = 0.04, respectively). The EF was comparable in both groups. There were no deaths and only one recurrent hospitalization due to HF during the study in a patient from the LVR group.

### Left-Ventricular Remodelling

Figure 2 shows the concentrations of EVs after AMI in patients with and without LVR at 6 months. The concentrations of EVs from endothelial cells (Figure 2A), erythrocytes (Figure 2C) and platelets (Figure 2E) were lower in patients who developed LVR, compared to those who did not develop LVR (*p* ≤ 0.02 for all), and discriminated between these two groups of patients (area under ROC curve [AUC] ≥0.74, *p* ≤ 0.02 for all) in univariate analysis (Figure 2B,D,F). The statistical estimates for prediction of LVR by EVs from endothelial cells, erythrocytes and platelets, including the cut-off values determined based on the ROC curves, are showed in Appendix A. The concentrations of other EVs, i.e., EVs from activated platelets, leukocytes and exposing PS, were comparable between the groups and did not predict LVR (Appendix A).

Because the concentrations of EVs from endothelial cells, erythrocytes and platelets discriminated between patients with and without LVR when measured separately, we incorporated these EV subtypes in a logistic regression model along with clinical data. To select the clinical data which should be incorporated in the model, we compared characteristics presented in Appendix A between patients with high and low concentrations of (i) endothelial cells, (ii) erythrocyte EVs and (iii) platelets EVs, based on the determined cut-off values. Patients with low endothelial EV concentrations (<3.64 × 10^5^ per ml) had higher INR (*p* = 0.024), compared to patients with high endothelial EV concentrations. Patients with low erythrocyte EV concentration (<1.67 × 10^7^ per ml) had higher creatinine (*p* = 0.041), compared to patients with high erythrocyte EV concentrations. Finally, patients with low platelet EV concentrations (<1.68 × 10^8^ per ml) had lower platelet count (*p* = 0.009), lower LDL-C concentration (*p* = 0.004) and higher prevalence of hypertension (*p* = 0.013) and smoking (*p* = 0.049), compared to patients with high platelet EV concentrations. Other clinical characteristics did not differ between the subsequent patient groups. Established predictors of LVR such as age, gender, AMI type and peak troponin I concentration were included in the model as well [24,25].

Multivariate logistic regression models for prediction of LVR was done for each EV subtype separately (Appendix A,). The concentrations of EVs from endothelial cells and erythrocytes were independent predictors of LVR (OR 8.2, CI 1.3–54.2, *p* = 0.03 for endothelial EVs; OR 17.8, CI 2.3–138.6, *p* = 0.01 for erythrocyte EVs). In contrast, EVs from platelets did not predict LVR in multivariate analysis (OR 21.5, CI 0.8–572.1, *p* = 0.07). However, combining the concentrations of EVs from endothelial cells, erythrocytes and platelets in one ROC curve improved LVR prediction (AUC 0.87, CI 0.73–1.00, *p* = 0.0004; Figure 3), compared to each EV subtype measured separately. No other clinical characteristics predicted LVR (Appendix A,).

## 4. Discussion

The main finding of our study is that the plasma concentrations of EVs from endothelial cells (CD146^+^), erythrocytes (CD235a^+^) and platelets (CD61^+^) 24 h after AMI predicted LVR at 6 months (Figure 2). Combining these 3 EV subtypes allowed to predict LVR after AMI with 83% sensitivity and 87% specificity (Figure 3).

Despite growing evidence demonstrating the role of EVs in cardiac and vascular remodelling [26], to the best of our knowledge this is the first clinical study to propose a novel strategy to predict LVR based on measuring the concentrations of EVs in plasma. Multiple other biomarkers including NT-proBNP, cardiac troponins, aspartate and alanine transaminase and C-reactive protein were evaluated for the prediction of LVR [27]. However, the combination of these biomarkers exhibited lower sensitivity and specificity compared to the strategy which we developed in this study, based on EVs [27]. In a recent systematic review summarizing the association between circulating biomarkers and LVR after AMI, 112 relations between 52 different biomarkers and LVR were reported [28]. The biomarkers most consistently associated with LVR included matrix metalloproteinase-9, collagen peptides, and B-type natriuretic peptide [28]. However, none of these biomarkers has been hitherto applied in a routine clinical setting to predict LVR due to their low sensitivity and/or specificity.

There is increasing evidence that EVs mediate the complex interplay between cardiomyocytes, fibroblasts, endothelial cells, vascular smooth muscle cells and extracellular matrix underlying LVR [29]. Depending on the cellular origin and concentration of EVs, EVs are either cardioprotective or promote adverse LVR [30]. In our study, decreased concentrations of EVs from endothelial cells (CD146^+^), erythrocytes (CD235a^+^) and platelets (CD61^+^) were associated with LVR.

Communication between endothelial cells and cardiomyocytes regulates cardiomyocyte function and the contractile state by providing both oxygenated blood supply and local protective signals that promote cardiomyocyte organization and survival [31]. For example, endothelial EVs transfer several active molecules including regulatory peptides and growth factors involved in angiogenesis and tissue reparation, therefore triggering epigenetic changes in the cardiomyocytes [26]. On the other hand, endothelial EVs might be involved in myocardial and vascular damage [26]. In our study, decreased concentrations of endothelial EVs were independent predictors of LVR, implying their cardioprotective properties, previously showed in cell cultures and animal models [32]. However, the population of endothelial EVs is very heterogeneous and their function is determined by the surface molecules and content [33]. Since we did not perform any functional experiments, we may only speculate about the role of CD146^+^ endothelial EVs in LVR.

Although erythrocytes are traditionally perceived as transporters of oxygen and nutrients to the tissues, recent experimental evidence indicates that they also participate in the nitric oxide metabolism and redox balance [34]. Myocardial reperfusion in patients with AMI decreased the activity of erythrocyte anti-oxidant enzymes, suggesting impaired anti-oxidant mechanisms after AMI [35]. Oxidative stress, in turn, plays an important role in HF pathophysiology. In a murine model, overexpression of the anti-oxidative enzyme glutathione peroxidase could attenuate post-AMI LVR and HF development [36]. Erythrocyte EV formation was showed to enable the selective removal of the oxidized proteins from erythrocytes [37]. In a transgenic murine model, erythrocyte EVs facilitated the cross-talk between erythrocytes and cardiomyocytes that contributed to the homeostasis after myocardial infarction [38]. Based on our results, it could be hypothesized that the decreased concentrations of erythrocyte EVs after AMI may indicate the disturbed erythrocyte redox balance, which contributes to LVR. Possibly, the degree of erythrocyte redox balance impairment might be associated with infarct area size. However, in our multivariate logistic regression model, AMI type and peak troponin I were not independent predictors of LVR. Hence, the factors responsible for erythrocyte redox balance and EV release after AMI remain to be established.

Platelets are widely recognized as key players in primary hemostasis and thrombosis. However, increasing experimental and clinical evidence shows that platelets contribute to many other pathophysiological processes including wound healing and cardiac regeneration through the release of growth factors, cytokines, and EVs [39,40]. Platelet EVs were showed to have pro-inflammatory properties in multiple studies. However, an initial, acute period of controlled inflammation may have a paradoxically beneficial role in cardiac recovery after AMI [41]. Whereas the direct interactions between platelet EVs and cardiomyocytes where not yet extensively studies, platelet EVs were demonstrated to improve endothelial cell function by decreasing endothelial permeability after thrombin challenge [42]. Finally, platelet EVs have been implicated in the therapeutic activity, of platelet-rich-plasma [43]. If so, EVs might be used not only to predict LVR, but also to augment regeneration of the post-infarct myocardium [44].

We did not use the second detection technique to study EVs, which does not comply with the MISEV 2018 (Minimal information for studies of extracellular vesicles) recommendations, which is a limitation of this study [45]. However, the goal of our study required both (i) to determine the EVs concentrations is a high-throughput way, and (ii) to determine the EVs cellular origin. None of the other well-established methods to study including nanoparticle tracking analysis, tunable resistive pulse sensing, Western blot or transmission electron microscopy would allow to fulfil both of these goals [46] and therefore they were not applied in this study. Another limitation is that we were not able to detect all EVs in plasma, since the EV-dedicated flow cytometer applied in this study has a detection limit of 150–200 nm for EVs [47], and most EVs have a diameter of less than 300 nm [48]. Hence, our findings specifically refer to EVs above the detection limit of the applied flow cytometer and cannot be extrapolated to the entire EV population. To increase the reliability of our findings and prove that the observed effects are really due to EVs and not due to other particles present in plasma (for examples chylomicrons), we compared the concentrations and predictive value of total particles and non-EV particles including chylomicrons (defined based on the differences in refractive index) [19] in patients with and without LVR (Appendix A). Neither total particles nor non-EV particles differed between patient groups and had predictive value for LVR, confirming that our findings are specifically due to EVs. Nevertheless, since we have not ultimately proved the association between the decreased concentrations of EVs and hard clinical end-points including recurrent hospitalizations or death due to HF, our results remain hypothesis-generating and required confirmation in future trials.

## 5. Conclusions

Decreased plasma concentrations of EVs from endothelial cells, erythrocytes and platelets predict LVR after AMI. Since EV release EVs contributes to cellular homeostasis by waste removal, decreased concentrations of EVs may indicate dysfunctional cardiac homeostasis after AMI, thus promoting LVR. Understanding how the communication between endothelial cells, erythrocytes and platelets cardiomyocytes is critical for cardiac regeneration after AMI.

## Figures and Tables

**Figure 1 biology-10-00097-f001:**
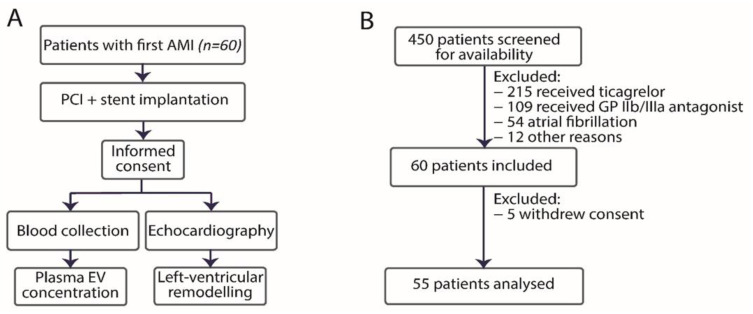
Study design (**A**) and inclusion and exclusion chart (**B**) Abbreviations: AMI–acute myocardial infarction; ASA–acetylsalicylic acid; GP–glycoprotein; PCI–percutaneous coronary intervention.

**Figure 2 biology-10-00097-f002:**
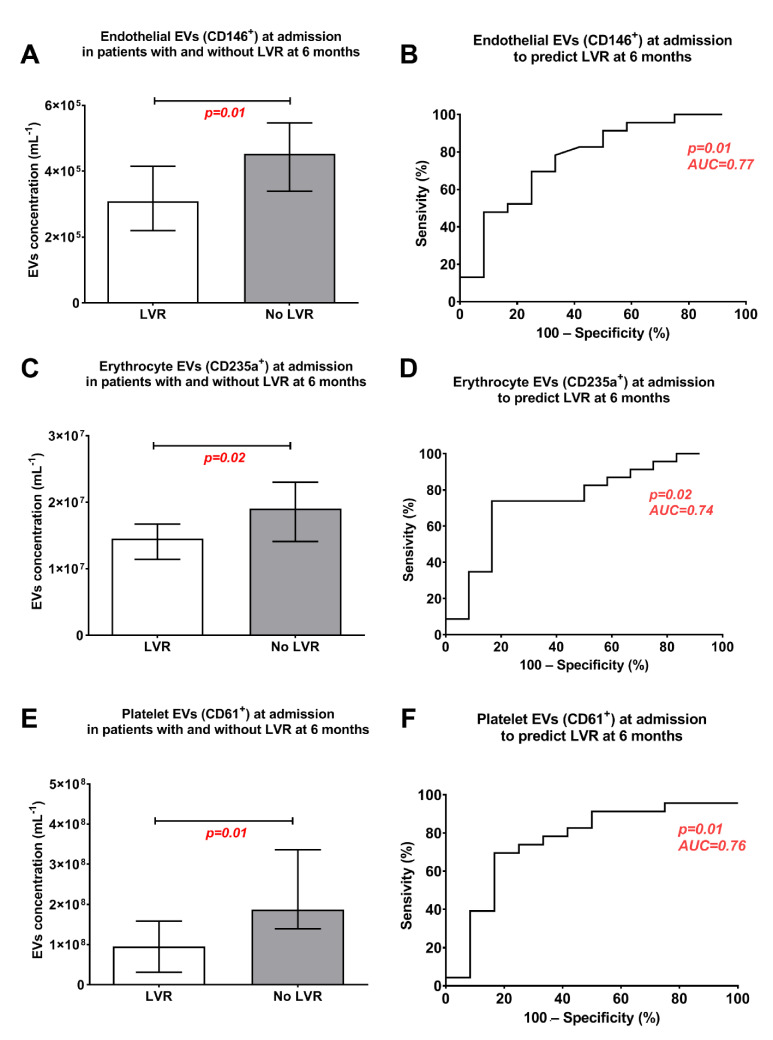
Concentrations of extracellular vesicles (EVs) measured with flow cytometry in platelet-depleted plasma prepared from patients with and without left ventricle remodelling (LVR) 6 months after AMI. (**A**,**B**): EVs from endothelial cells. (**C**,**D**): EVs from erythrocytes. (**E**,**F**): EVs from platelets.

**Figure 3 biology-10-00097-f003:**
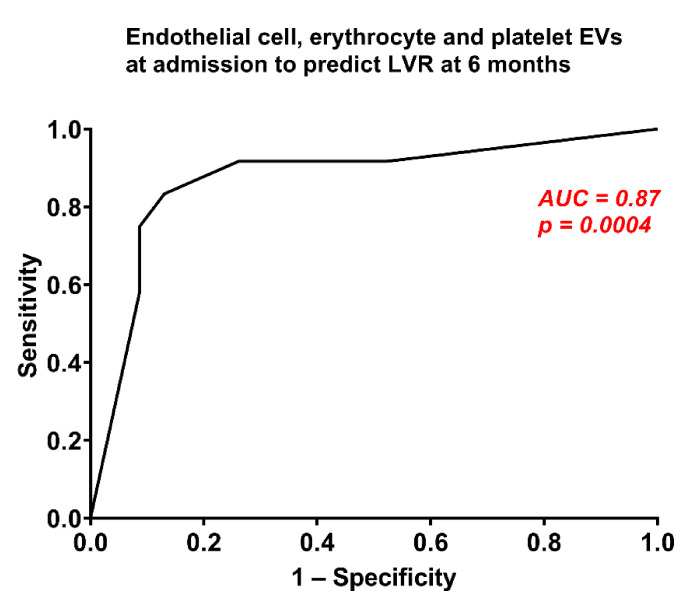
Combination of the concentrations of extracellular vesicles (EVs) from endothelial cells, erythrocytes and platelets based on the cut-offs to predict left ventricular remodelling (LVR) after AMI.

**Table 1 biology-10-00097-t001:** Main characteristics of patients with and without left ventricular remodelling (LVR) at 6 months after acute myocardial infarction.

Characteristic	LVR (*n* = 12)		No LVR (*n* = 43)		*p*-Value
	N	SD, range, %	N	SD, range, %	
Age, years–mean ± SD	59.5	10.3	65.6	9.3	0.06
Male gender–number (%)	8	67	32	74	0.72
BMI–median (IQR)	28.1	25.5–29.9	27.9	25.4–31.8	0.73
Diagnosis at admission–number (%)					
STEMI	11	92	31	72	0.26
Anterior AMI	2	17	8	19	1.00
Cardiovascular risk factors–number (%)					
Arterial hypertension	8	67	26	60	0.75
Diabetes mellitus	3	35	7	16	0.67
Dyslipidaemia	8	67	29	67	1.00
Smoking	5	41	18	41	1.00
Laboratory characteristics at baseline					
Creatinine, mg/dL–median (IQR)	0.91	0.73–1.02	0.95	0.81–1.05	0.39
C-reactive protein–median (IQR)	3.75	1.73–6.08	3.00	1.7–5.9	0.16
Haemoglobin, g/dL–mean ± SD	14.0	1.3	13.9	1.3	0.89
INR–mean ± SD	1.14	0.16	1.07	0.08	0.06
NT-proBNP–median (IQR)	696	386–1936	888	192–1978	0.49
Platelet count, 10^3^/μL–mean (SD)	262.3	75.4	235.6	69.2	0.27
Troponin I, ng/mL–median (IQR)	16.2	5.2–42.6	14.7	2.5–35.5	0.46
Echocardiography at baseline					
LVEDV, mL–median (IQR)	104	92–120	105	95–123	0.58
LVESV, mL–median (IQR)	41	38–52	41	40–65	0.38
LVEF, mL–median (IQR)	53	47-57	51	45–54	0.43
Echocardiography at 6 months					
LVEDV, mL–median (IQR)	107	97–126	83	62–93	0.03
LVESV, mL–median (IQR)	59	43–63	57	27–48	0.04
LVEF, mL–median (IQR)	56	50–58	60	52–63	0.10
Pharmacotherapy at discharge–number (%)					
Aspirin	12	100	43	100	1.00
P2Y12 inhibitor	12	100	43	100	1.00
Atorvastatin	11	92	42	98	0.39
β-blocker	10	83	39	90	0.60
ACE-inhibitor or ARB	11	92	41	95	0.53
Aldosterone receptor antagonist	3	25	11	26	1.00
Proton pump inhibitor	11	92	40	93	1.00
Pharmacotherapy at follow-up–number (%)					
Aspirin	12	100	43	100	1.00
P2Y12 inhibitor	12	92	43	100	1.00
Atorvastatin	11	92	41	95	0.53
β-blocker	10	83	38	88	0.64
ACE-inhibitor or ARB	11	92	42	98	0.40
Aldosterone receptor antagonist	4	33	11	26	0.72
Proton pump inhibitor	10	83	41	95	0.20

Abbreviations: ACE: angiotensin-converting enzyme; ARB: angiotensin-receptor blockers; BMI: body mass index, weight in kilograms divided by square of the height in meters; CK-MB: creatine kinase muscle-brain isoenzyme; CVD: cardiovascular disease; GLS: global longitudinal strain; INR: international normalized ratio; IQR: interquartile range; LDL-C: low-density lipoprotein-cholesterol; LVEDD: left ventricle end-diastolic diameter; LVEDV: left ventricle end-diastolic volume; LVESV: left ventricle end-systolic volume; LVEF: left ventricle ejection fraction; NSTEMI: non-ST-segment elevation myocardial infarction; NT-proBNP: N-terminal pro-B-type natriuretic peptide; SD: standard deviation; STEMI: ST-segment elevation myocardial infarction.

## Data Availability

Source data including flow cytometry files are available upon request.

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
