# Peer review of "Plasma Concentrations of Extracellular Vesicles Are Decreased in Patients with Post-Infarct Cardiac Remodelling"

_biology, 2021, doi:10.3390/biology10020097_

Round 1
Reviewer 1 Report
This is an interesting and well-written paper showing the potential value of measuring RBC-, platelet-, and endothelial-derived extracellular vesicles in predicting left ventricular remodeling. The methodology is convincing and AUC of 0.87 (with sensitivity of 83% and specificity of 87%) is also promising. However, the Discussion must be further elaborated and reinforced by the papers demostrating the positive effects of RBC-, platelet, and endothelial-derived extracellular vesicles on heart physiology. Have there been any studies showing a mechanistical explanation of reduced EV production upon LVR-related pathological conditions?
It would be also good if the authors describe advantages and pitfalls of flow cytometry-based measurement of extracellular vesicles in the human serum including optimised flow rates to exclude swarm detection. In my laboratory practice, swarm detection significantly complicates the reliability of the approach as many flow cytometers do not allow flow rate of 3 μL/min.
The articles can be accepted for publication upon a substantial revision of the Discussion section.
Author Response
Dear Reviewer,
Thank you a lot for your help to improve our manuscript. Please find our response attached.
Best regards,
Aleksandra Gąsecka

Reviewer 2 Report
The objective of this study has been to assess whether circulating levels of extracellular vesicles (EV) from various cell sources could serve as predictive biomarkers of left ventricular remodeling in patients revascularized after an acute myocardial infarction. The authors have studied 55 patients and measured plasma concentrations of EV from endothelial cells (CD146+), erythrocytes (CD235a+), leukocytes (CD45+), platelets (CD61+) and activated platelets (P-selectin+) by flow cytometry using a device which allows to detect small particles. Left ventricular remodeling was defined as an increase in left ventricular end-diastolic volume by 20% at 6 months after infarction, compared to baseline. The major finding was that a decrease in plasma concentrations of EV from endothelial cells, erythrocytes and platelets predicted remodeling in univariate analysis. However, using multivariate logistic regression models, it was found that only EV from endothelial cells and erythrocytes were independent predictors of remodeling although combining the concentrations of EV from endothelial cells, erythrocytes and platelets in one ROC curve improved the prediction of remodeling with an AUC of 0.87.
This study addresses a clinically relevant topic as there is a continued quest for biomarkers predictive of remodeling. The results look straightfroward but the major issue is the incomplete characterization of the EV. This is obviously a crucial point since the paper is exclusively focused on their detection and one knows that such a detection is fraught with several technical pitfalls. Thus, the data cannot be presented as representative of EV in the absence of a more thorough assessment of the particles which should at least include Nanoparticle Tracking Analysis and additional Western blots analyses looking at some specific EV-associated markers like CD63 or CD81.The "Methods" section of the paper should thus comply with the MISEV recommandations as published in the paper by Thery et al. (J Extracell Vesicles. 2018 Nov 23;7(1):1535750. ) which should be quoted.
Furthermore, given the spatial resolution of the flow cytometer which has been used, only particles > 200 nm can be detected and it should thus be specified that the EV primarily reperesented microparticles with the likely exclusion of exosomes.
It would also be informative to know whether the differences in remodeling, as assessed by echocardiography, were paralleled by differences in the clinical outcomes of the patients (primarily rehospitalization for heart failure - the authors mention one case without indicating to which group the patient belonged - but also clinical symptoms and/or BNP levels) and to know whether the treatment regimens implemented during the 6-month follow-up was consistent in all patients. The authors provide the list of medications at the time of discharge but do not give information as to whether there was subsequently a similar compliance to the drugs between patients who developed remodeling and those who did not.
Finally, the authors should at least discuss some mechanistic hypotheses allowing to relate the decrease in the plasma concentrations of EV from specific cell sources to the limited development of left ventricular dilation over time. The explanation provided at the end of the discussion is rather vague. The study should have included in vitro potency assays allowing to assess the more specific effects of the EV from the diverse cellular sources (erythocytes, leukocytes, platelets) on some critical end points like cell survival or apoptosis as this might have provided some clues about the reasons why a decrease in some of these EV subsets had long term deleterious effects on heart geometry.
Author Response
Dear Reviewer,
we are thankful for the time and effort that you spent to provide in-depth review of our manuscript. We corrected our manuscript according to your suggestions. Our response and corrections are listed in the attached file.
Best regards,
Aleksandra Gasecka

Round 2
Reviewer 2 Report
The authors have tried to scholarly address the issues that had been raised and their answers are satisfactory.
The last sentence of the Discussion "Understanding how the communication between en-dothelial cells, erythrocytes and platelets cardiomyocytes is critical for cardiac regenera-tion after AMI." just needs to be reworded as it looks grammatically incomplete.
I think that the relevance of the topic as well as the improvements of the manuscript warrant its publication.